# Technical Implications for Surgical Resection in Locally Advanced Pancreatic Cancer

**DOI:** 10.3390/cancers15051509

**Published:** 2023-02-28

**Authors:** Martín de Santibañes, Juan Pekolj, Rodrigo Sanchez Claria, Eduardo de Santibañes, Oscar Maria Mazza

**Affiliations:** 1Liver Transplant Unit, Department of Surgery, Division of HPB Surgery, Hospital Italiano de Buenos Aires, Juan D. Perón 4190, Buenos Aires C1181ACH, Argentina; 2Department of Surgery, Division of HPB Surgery, Hospital Italiano de Buenos Aires, Buenos Aires C1181ACH, Argentina

**Keywords:** locally advanced pancreatic ductal adenocarcinoma, neoadjuvant treatment, extended pancreatectomies, portomesenteric venous resection, arterial resection

## Abstract

**Simple Summary:**

Neoadjuvant treatment followed by highly complex surgical procedures has been studied over the last decade with promising short- and long-term results in patients with locally advanced pancreatic ductal adenocarcinoma (LAPC). In recent years, a wide variety of complex surgical techniques that involve extended pancreatectomies, including portomesenteric venous resection, arterial resection, or multi-organ resection, have emerged to optimize local control of the disease and improve postoperative outcomes. We aim to describe the preoperative surgical planning as well different surgical resections strategies in LAPC after neoadjuvant treatment in an integrated way for selected patients with no other potentially curative option other than surgery.

**Abstract:**

Pancreatic ductal adenocarcinoma remains a global health challenge and is predicted to soon become the second leading cause of cancer death in developed countries. Currently, surgical resection in combination with systemic chemotherapy offers the only chance of cure or long-term survival. However, only 20% of cases are diagnosed with anatomically resectable disease. Neoadjuvant treatment followed by highly complex surgical procedures has been studied over the last decade with promising short- and long-term results in patients with locally advanced pancreatic ductal adenocarcinoma (LAPC). In recent years, a wide variety of complex surgical techniques that involve extended pancreatectomies, including portomesenteric venous resection, arterial resection, or multi-organ resection, have emerged to optimize local control of the disease and improve postoperative outcomes. Although there are multiple surgical techniques described in the literature to improve outcomes in LAPC, the comprehensive view of these strategies remains underdeveloped. We aim to describe the preoperative surgical planning as well different surgical resections strategies in LAPC after neoadjuvant treatment in an integrated way for selected patients with no other potentially curative option other than surgery.

## 1. Introduction

Pancreatic ductal adenocarcinoma (PDAC) remains a global health challenge and is predicted to soon become the second leading cause of cancer death in developed countries [1]. Currently, surgical resection in combination with systemic chemotherapy offers the only chance of cure or long-term survival. However, only 20% of cases are diagnosed with anatomically resectable disease. Even localized PDAC should be approached as a systemic disease. For patients with locally advanced pancreatic ductal adenocarcinoma (LAPC), neoadjuvant treatment (chemotherapy with or without chemoradiotherapy) followed by highly complex surgical procedures has been studied during the last decade with promising short- and long-term results [2].

In recent years, a wide variety of complex surgical techniques that involve extended pancreatectomies, including portomesenteric venous resection (PVR), arterial resection (AR), or multi-organ resection, have emerged to optimize local control of the disease and improve postoperative outcomes. Microscopic tumor involvement in the resection margin and lymph node metastases are common in this scenario, and, therefore, local recurrence is frequent and conditions patient survival. One of the main challenges in LAPC is achieving tumor-free resection margins (R0 = tumor-free margin > 1 mm) [3] and lymph nodes clearance due to extensive tumor burden and dense desmoplastic tissue. Although there are multiple surgical techniques described in the literature to improve outcomes in LAPC, the comprehensive view of these strategies remains underdeveloped. We aim to describe the preoperative surgical planning as well different surgical resections strategies in LAPC after neoadjuvant treatment in an integrated way for selected patients with no other potentially curative option other than surgery.

## 2. Preoperative Surgical Planning

Multidetector computed tomography (MDCT), using specific pancreatic contrast protocols, represents the most widespread method for establishing a suspected diagnosis of PDAC. It allows precise staging of the disease, determining tumor resectability and surgical planning according to vascular variations and/or adjacent organ invasion. Historically, localized pancreatic disease has been classified as resectable (without vascular involvement by imaging methods) or locally advanced (unresectable, with extensive arterial or venous vascular involvement). According to the consensus statement of the International Study Group of Pancreatic Surgery (ISGPS) [4], which is based primarily on the recommendations of the National Comprehensive Cancer Network (NCCN), LAPC presents involvement of the superior mesenteric artery (SMA) or the celiac trunk (CT) in more than 180° of the vascular circumference or compromise of the aorta and/or compromise of the superior mesenteric vein (SMV) or portal vein (PV), which makes it impossible to provide adequate vascular resection and reconstruction in the absence of distant metastatic disease (Figure 1). The term borderline has been used to describe tumors that are potentially resectable, but that have some degree of vascular involvement. A borderline tumor would be one with reconstructable venous involvement (SMV or PV) and/or contact within 180° of the vascular circumferences of arterial structures.

However, considerable disparities in multidisciplinary team evaluations of patients with pancreatic cancer exist, including substantial variation in resectability assessments [5]. A recent symposium of experts from Western and Eastern high-volume centers reported new resectability classifications from their respective institutions based on tumor biology, conditional status, pathology, and genetics, in addition to anatomical tumor involvement. Interestingly, experts from all the centers reached the agreement that anatomy alone is insufficient to define resectability in the current era of effective neoadjuvant therapy [6]. Neoadjuvant chemotherapy, with or without chemoradiotherapy, may result in successful resection in up to 60% of patients with LAPC with a substantial survival advantage [7]. In high-volume pancreatic centers and after discussion by a multidisciplinary team, surgical exploration may be suggested in patients with non-progressive (stable or regressing) RECIST criteria [8] to assess the possibility of pancreatic resection. Nevertheless, as the resectability of PDAC is well-defined by vascular involvement rather than tumor volume, RECIST is not suitable for the evaluation of tumor response following neoadjuvant treatment. Moreover, MDCT may underestimate the response of neoadjuvant therapy and, therefore, the discrimination of the venous and/or arterial compromise, since the discrimination between fibrosis and viable tumor remains very complex. A recent development in post-process-rendering, called cinematic rendering, overcomes this by utilizing advanced light modeling to generate photorealistic 3D images with enhanced details. For local determination of resectability, vascular mapping allows for accurate assessment of major arteries and the portovenous system. For the portovenous anatomy, it assists in determining the optimal surgical approach (extent of resection, appropriate technique for reconstruction, and need for mesocaval shunting). For arterial anatomy, vessel encasement either represents dissectible involvement via periadventitial dissection or true vessel invasion that is unresectable [9]. Magnetic resonance imaging—halo sign, defined as replacement of solid perivascular (arterial and venous) tumor tissue by a zone of fatty-like signal intensity—might be helpful to assess the effect of induction chemotherapy in patients with LAPC [10]. Further investigations incorporating quantitative parameters such as radiomics and deep learning may improve diagnostic performance of MDCT for predicting R0 resection [11].

Another important aspect during patient work-up is the decreased levels of tumor marker serum carbohydrate antigen (CA) 19-9 after neoadjuvant therapy, because this may predict a better prognosis, with low incidence of hepatic recurrence after surgery [12]. Rose et al. [13] identified that the percent decrease in CA19-9 from baseline to minimum value (odds ratio [OR] 0.947, *p* ≤ 0.0001) and the percent increase from minimum value to final restaging CA19-9 (OR 1.030, *p* ≤ 0.0001) were predictive of tumor progression in patients with advanced pancreas cancer.

Tanaka et al. [14] recently described that the shrinkage rate of the primary tumor, the response rate of MDCT density attenuation of the tumor, and post-chemotherapy CA19-9 serum levels were independent predictors of survival in patients with resected LAPC after preoperative treatment with FOLFIRINOX. Then, 18 F-fluorodeoxyglucose PET/CT was proposed as a radiologic marker to predict the prognosis and treatment response of neoadjuvant therapy for PDAC [15]. Recently, Abdelrahman et al. [16] showed that among patients with post-neoadjuvant therapy, FDG PET highly predicts pathologic response (odds ratio, 43.2; 95% CI, 16.9–153.2), recurrence-free survival (hazard ratio, 0.37; 95% CI, 0.2–0.6), and overall survival (hazard ratio, 0.21; 95% CI, 0.1–0.4), and is superior to biochemical responses alone (CA 19-9). 

A recent classification proposed a four-stage Whipple procedure categorization based on the extent of surgery and surgical outcomes. Multivisceral pancreatoduodenectomy (type 3) or pancreatoduodenectomy with arterial resection (type 4) had increased probability of surgical complications, relaparotomy, and 90 day mortality [17]. Type 3 and 4-types are correlated with pancreatic resections in LAPC. Therefore, thorough extensive preoperative work-up stratification, taking into account age, comorbidities, functional status, and the viability of the procedure can improve patient selection and predict adverse postoperative events in major cancer surgery [18]. We select our LAPC patients for surgery after neoadjuvant therapy, considering many of these anatomical and biologic criteria and conditional parameters described above, discussed on a case-by-case basis in our multidisciplinary committee.

## 3. Surgical Aspects

The early steps of the operation are similar as in conventional pancreatic resection and some technical aspects have been described previously [19]. However, we would like to emphasize some aspects that may be essential to resolve complex vascular resections and achieve local and overall recurrence. 

Through an extensive Kocher maneuver in combination with the Cattell–Braasch mobilizations of the cecum, right colon, right colonic flexure, and the root of the small bowel. together with the Treitz ligament. an adequate exposure of the entire infrahepatic vena cava (IVC), aorta, the left renal vein (LRV), and the right origin of the SMA, which is situated just above the LRV, can be achieved and offer adequate tractability of the mesenteric root in case segmental vein resection and reconstruction should become necessary [20] (Figure 2). The origin of the SMA can be marked by a vessel loop to recognize the vessel during the margin dissection and have the facility to clamp the SMA if needed, to prevent bowel congestion during complex venous reconstructions. In the literature there are different approaches to the SMA [21]. All alternatives of the artery-first techniques have in common the fact that dissection is achieved within the tunica adventitia of the SMA. The approach depends on the results of preoperative imaging defining the place of the most likely tumor relating to the vessel. The SMA can be approached from a left-sided infracolic approach if tumors of the body or tail of the pancreas are supposed to infiltrate the artery from this direction [22]. The small bowel can be flipped to the right side of the patient, and the peritoneum opened along the mesentery root parallel and to the left of the proximal jejunum and the duodenojejunal flexure. The origin of the SMA from the right was already identified in the angle formed by the IVC and LRV with the extended Kocher maneuver; on the right side of the SMA, a replaced or accessory right hepatic artery, if existing, can be identified and preserved and the dissection is carried out cephalad beside the aorta until the origin of the SMA is reached. 

In the situation that MDCT shows tumor extension close to the jejunal branches of the SMV or distal aspects of the SMA, the identification of the superior mesenteric vessels is performed at the root of the mesentery below the transverse mesocolon to the origin from the aorta. The middle colic artery is detached at its origin from the SMA in this manner, and the tumor-infiltrated section of the transverse mesocolon is resected to remain with the specimen (Figure 3). The first jejunal loop is consequently divided and moved to the superior right side of the abdomen. Meticulous dissection beside the SMA with alternation between both directions of dissection, leaves the right lateral circumference of the SMA free from all adjacent soft tissue with the inferior pancreaticoduodenal vessels severed at their origin, or even more clearing the autonomous nerves from the right and posterior circumference of the SMA [17,19,23].

A recent systematic review and meta-analysis showed that patients undergoing an artery-first approach to pancreatoduodenectomy may be associated with improved perioperative outcomes and survival in comparison with those having standard pancreatoduodenectomy [24]. The strategy depended on the results of preoperative imaging defining the site of the most likely tumor infiltration. 

Standard distal pancreatectomy and splenectomy for PDAC in the body or tail have been associated with high positive margin rates and poor overall survival in relation to tumor infiltration of the anterior renal fascia and the left adrenal gland [25]. For this purpose, radical antegrade modular pancreatosplenectomy (RAMPS) is suggested [26]. Conventional RAMPS proceeds in a right-to-left antegrade manner, with early parenchymal transection at the neck of the pancreas and early control of the splenic vessels (in its origin), CT and SMA lymphadenectomy, as well as full visualization of the retroperitoneal plane of dissection (Figure 4A). The posterior magnitude of dissection can result in front of the adrenal gland, behind the anterior renal fascia (anterior RAMPS), or behind the left adrenal gland (posterior RAMPS) (Figure 4B). The primary goals of RAMPS are to increase the rate of R0 resection and lymph node yield for pancreatic cancer in the body or tail. The resolution to perform anterior or posterior RAMPS is made based on the posterior extent of tumor invasion.

It has been theorized that one of the explanations for the poor long-term survival is that tumor-infiltrated autonomous nerve spreads frequently in the preaortic region and this spread can lead to positive resection margins or sites of tumor-infiltrated lymphatic tissue.

The TRIANGLE operation [27] is a proper approach to achieve a complete and radical removal of the tumor and associated lymphatic or perineural extension along a region defined anatomically by the origins of the CT (superiorly), the SMA (inferiorly), and the portal vein (anteriorly) (Figure 5). The dissection of the triangle region is best conveyed after the pancreatic head has been entirely mobilized from the SMA and follows the CT from its origin to the common hepatic artery. The CT and SMA might be totally dissected from the right (pancreatoduodenectomy) or the left side (distal pancreatosplenectomy). If a total pancreatectomy has been executed, both arterial vessels should be dissected circumferentially.

Extended resection of neural and lymphatic tissue carries a risk of increased surgical morbidity, including adverse effects such as postoperative bleeding, uncontrolled diarrhea, and ascites.

## 4. Extended Pancreatectomy

As mentioned above, the LAPC clinical scenario can be associated with unconventional resections of organs adjacent to the pancreas that involve multivisceral resections. The ISGPS published a list of structures and organs additionally to the ones resected in a standard pancreatoduodenectomy, pancreatic left resection, or a total pancreatectomy because of the many different classifications that existed to that point [28]. Some publications propose that the surgical morbidity is increased in extended resections while overall perioperative mortality appears to be similar compared with standard pancreatectomies [29]. This increased perioperative morbidity demands close postoperative follow-up of these patients and an elaborate and aggressive management of complications that can only be provided in specialized high-volume centers.

## 5. Vein Resection and Reconstruction

Pancreatic surgery in combination with PV and/or SMV resection represent a frequent and more complex surgical scenario in patients with LAPC compared with standard pancreatic resection. Although several studies suggest that pancreatic resections with PVR are associated with acceptable perioperative risk, performing venous resection undoubtedly adds a technical challenge to an already complex surgical procedure [30]. The benchmark cohort revealed a 4% or less in-hospital mortality, with a portal vein thrombosis rate ≤ 14% [30]. A nationwide cohort analysis showed that patients with segmental resection, but not those who had wedge resection, had higher rates of major morbidity (odds ratio = 1.93, 95% CI 1.20 to 3.11) and worse overall survival (hazard ratio = 1.40, 95% CI 1.10 to 1.78) compared to patients without venous resection [31].

The extent of en bloc venous resection is related to the possibility of venous reconstruction, while the technique of vascular reconstruction differs drastically based on anatomical vascular variations and surgeon preferences. The ISGPS suggests a specific categorization of the types of venous reconstruction to be incorporated in analyses for more detailed and evidence-based evaluation in patients with venous involvement [4]. Small venous wedge resections can be resolved with direct suture of the vein (type 1). In this circumstance, all types of venous narrowing should be avoided to prevent thrombotic complications (Figure 4A).

A lateral patch into the venous defect can be safely used for some defects (type 2). Autologous substitute for venous reconstruction, such as parietal peritoneum, can be harvested from the diaphragm, the right or left hypochondrium, or the falciform ligament [32] (Figure 6A). The mesothelial layer of the patch can be placed on the intraluminal side of the vein and the musculoaponeurotic layer outside (Figure 6B). These autologous sources represent a quick and accessible alternative for vascular reconstruction, especially when the need for venous resection is unexpected.

Segmental defects can frequently be reconstructed with primary end-to-end anastomosis (type 3) (Figure 7). With the Cattell–Braasch mobilizations of the right hemicolon and mesenteric root, together with the Treitz ligament, an appropriate flexibility for segmented vascular resection and tension-free anastomosis of the porto–mesenteric axis can be obtained. If the splenic vein needs to be divided, various possibilities exist. In most patients, the splenic vein can be ligated, without clinical intervention. However, if venous congestion of the stomach or spleen occurs, the splenic vein can be reimplanted into the PV or SMV in an end-to-side fashion. 

The use of a vascular graft interposition (type 4) is contemplated for the reconstruction of large vascular segment defects (Figure 8). Suitable autologous graft substitutions for venous reconstruction include LRV, saphenous vein, inferior mesenteric vein, jugular vein, gonadal vessels, peritoneal substitutes (with the possibility of peritoneal tubular graft confection), cryopreserved veins, cadaveric graft veins, or synthetic graft prothesis with materials such as polytetrafluoroethylene (PTFE). The use of PTFE has the drawback of long-term anticoagulation in relation to high risk of vascular thrombosis or prothesis infection.

Cavernous transformation of the PV represents a challenging surgical scenario in LAPC. This situation may be associated with complete tumor occlusion of the PV/SMV or in relation to a paraneoplastic procoagulant disorder. The technique of “venous bypass graft first” approach was recently described to avoid major bleeding complications, intestine congestion, or liver perfusion disorders [33,34]. The procedure includes the identification of the SMV or one of its branches (in the mesenteric root) as well as the PV or vascular tributaries to the liver (pericholedochal varices) in the hepatoduodenal ligament, adjacent to the liver hilum. A jump graft between these vascular structures can be used for this purpose (autologous, cadaveric bank graft, or synthetic prothesis) (Figure 9). This surgical strategy offers continuous portal blood flow to the liver during the resection and reconstruction phase of the operation. Bachellier et al. [35] suggested that venous shunt seemed necessary only for patients with intra-abdominal collateral circulation (types C and D), which maintains the portal inflow by filling the PV downstream to the venous stenosis or occlusion.

## 6. Arterial Resection

Although recent meta-analyses concluded that pancreatectomy with AR were associated with increased morbidity and mortality in comparison to non-AR pancreatectomies [36,37], the introduction of new chemotherapy schemes (FOLFIRINOX or gemcitabine + nabpaclitaxel), have changed the paradigm of the treatment approach for LAPC in selected patients with arterial compromise (especially in young patients if a R0 situation can be achieved) [38]. Recently, Tee et al. [39] published the largest single-institution series specifically addressing indications, outcomes, and perioperative risk factors in pancreatectomy with AR. Despite having described a significant improvement in 90 day mortality over time, morbidity and the use of hospital resources remain unchanged. The most significant predictor of worse outcomes is post-pancreatectomy hemorrhage (PPH). Graft reconstruction and pancreatic fistula were also associated with increased major morbidity in their experience. It is highly recommended such cases be performed by surgeons with the specific anatomic comprehension and skillsets required not only to perform such complex resections, but also with the necessary institutional expertise and immediate availability of interventional radiology, complex endoscopy, and adequate intensive care facilities. On the other hand, PPH after pancreatectomy with AR may be a logical consequence of postoperative pancreatic fistula. To eliminate this risk, total pancreatectomy has been suggested [40]. However, a recent study found no protective effect of total pancreatectomy on its outcomes [41].

From the anatomical and technical point of view, there is a great difference between AR of the SMA, with respect to the CT and the hepatic artery. For example, after resection of the CT, the blood supply to the liver and pancreas head via the common hepatic artery relies on retrograde arterial perfusion of the pancreatoduodenal arcades and the gastroduodenal artery with the blood flow coming from the SMA. To improve collateral flow tributaries and reduce postoperative liver ischemia, we applied preoperative common hepatic artery embolization [42]. We had previously described some surgical strategies for restoring liver arterial perfusion in pancreatic resections [42]. In case of short-segment resection of the hepatic artery or SMA, reconstruction can occasionally be performed by direct end-to-end anastomosis. However, most cases of AR include longer vascular defects and complex surgical strategies to restore arterial perfusion, with any type of graft interposition or transposition (Figure 10). 

Recently, the technique of “arterial divestment” opens a great possibility for a periarterial neurolymphatic tissues dissection without the need of AR and respective reconstruction, with the risks that this implies. Arterial clearance is achieved by placing it in a plane between the uninvolved arterial wall and the tumor tissue of the affected arterial segment [43]. The dissection plane should be placed between the periarterial nerve plexus and the arterial adventitia (Figure 11).

## 7. Conclusions

In this review, recent surgical and technical aspects in LAPC were discussed, including extended pancreatic resections, PMV resections and their respective forms of vascular reconstruction, arterial resection, or alternatively arterial divestment. Furthermore, through multimodal treatment systems and appropriate surgical resection techniques, the long-term outcome after extended pancreatectomies can be similar to standard resections and considerably better than palliative care, despite the fact that we do not have the sufficient level of scientific evidence.

## Figures and Tables

**Figure 1 cancers-15-01509-f001:**
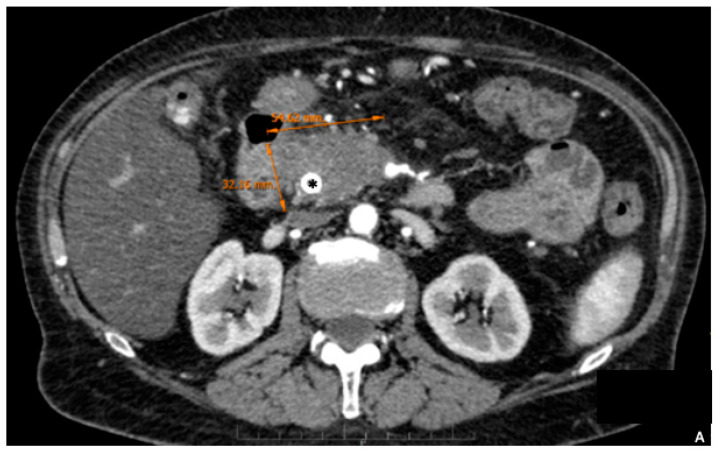
A 67 year old female patient with jaundice and abdominal pain. Endoscopic ultrasound biopsy confirmed poorly differentiated PDAC and a biliary stent was placed (asterisk). MDCT shows expansive pancreatic formation located in the cephalic portion, with poorly defined limits, measuring approximately 54 × 32 mm (**A**). Exophytic tumor (T), encompassing the SMA and distal branches (white arrow) and the spleno–mesenteric–portal confluence, extending towards the superior mesenteric vein (encasement) (**B**,**C**). Cephalically it extends towards the celiac trunk, encompassing its terminal branches (black arrow) (**D**).

**Figure 2 cancers-15-01509-f002:**
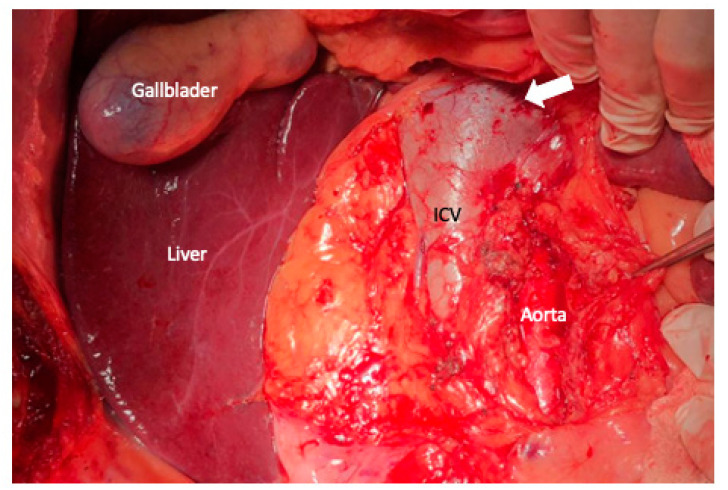
In extensive Kocher maneuver in combination with the Cattell–Braasch mobilizations of the cecum, right colon, right colonic flexure, and the root of the small bowel, together with the Treitz ligament, an adequate exposure of the entire infrahepatic vena cava (IVC), aorta, and the left renal vein (white arrow) is achieved.

**Figure 3 cancers-15-01509-f003:**
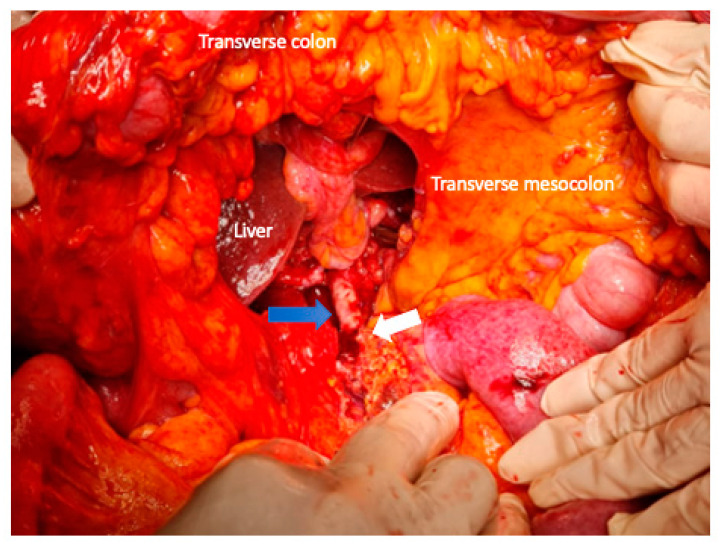
Identification of the superior mesenteric vessels (the blue arrow shows the superior mesenteric artery) is performed at the root of the mesentery below the transverse mesocolon to the origin from the aorta. The middle colic artery is detached at its origin from the SMA in this manner (white arrow), and the tumor-infiltrated section of the transverse mesocolon is resected to remain with the specimen.

**Figure 4 cancers-15-01509-f004:**
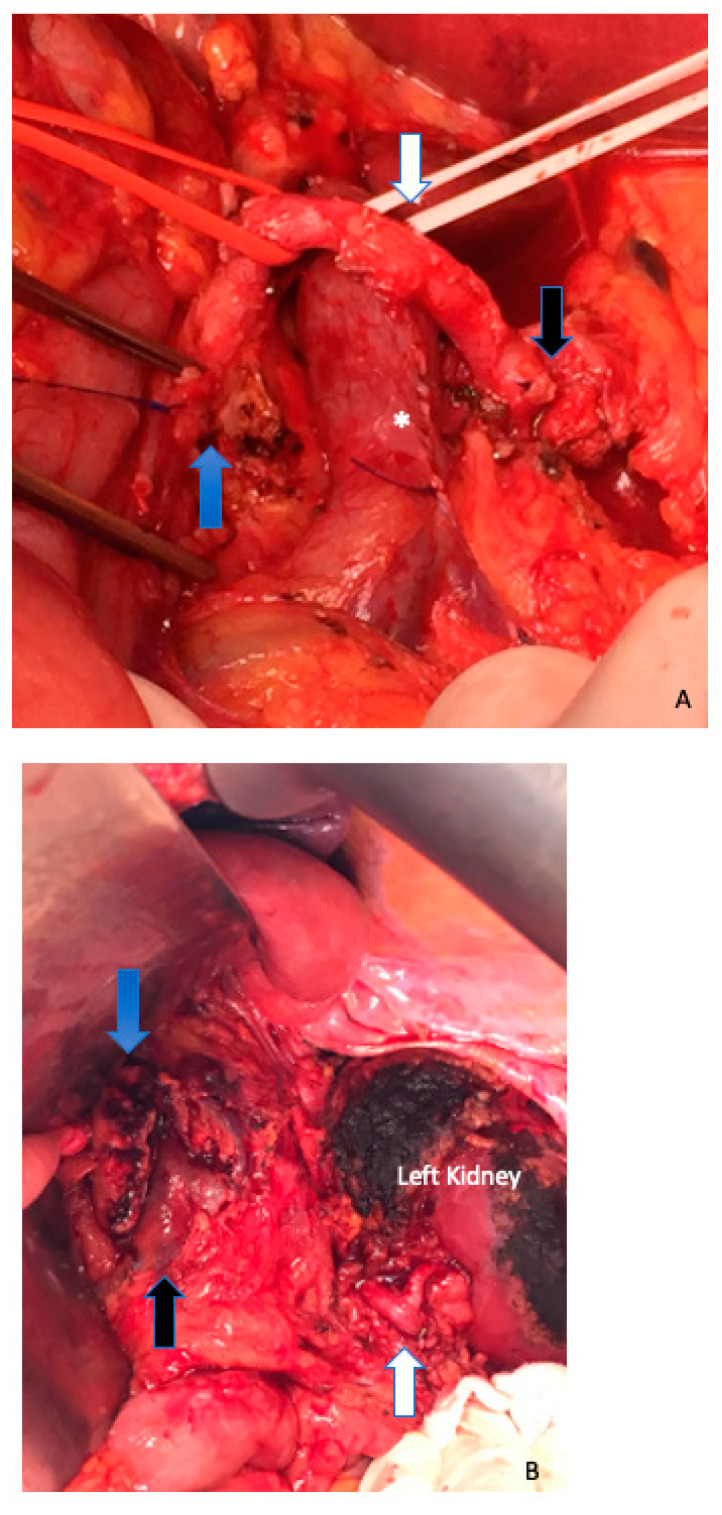
(**A**) Radical antegrade modular pancreatosplenectomy (RAMPS) proceeds in a right-to-left antegrade manner, with early parenchymal transection at the neck of the pancreas, close to the gastroduodenal artery (blue arrow) and early control of the splenic vessels (in its origin) (white arrow). The black arrow marks the common hepatic artery, and the asterisk shows a partial venous excision spleno–mesenteric confluence with direct closure (venorrhaphy). (**B**) Posterior RAMPS. The posterior magnitude of dissection was behind the left adrenal gland, including the Gerota fascia and the fat tissue around the left kidney. The left renal artery is shown with white arrow. Blue arrow shows the pancreas stump. Black arrow, superior mesenteric vein.

**Figure 5 cancers-15-01509-f005:**
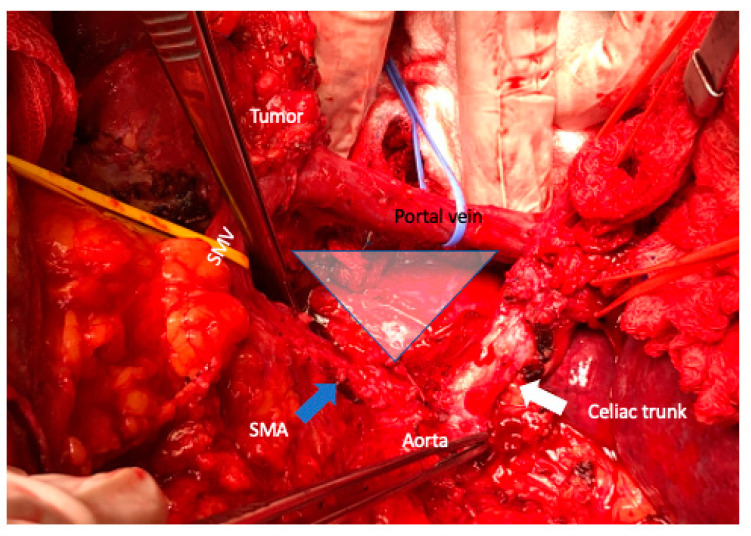
The TRIANGLE operation (blue triangle zone) is a proper approach to achieve a complete and radical removal of the tumor and associated lymphatic or perineural extension along a region defined anatomically by the origins of the celiac trunk (white arrow), the superior mesenteric artery (SMA) (blue arrow), the portal vein (anteriorly), and superior mesenteric vein (SMV).

**Figure 6 cancers-15-01509-f006:**
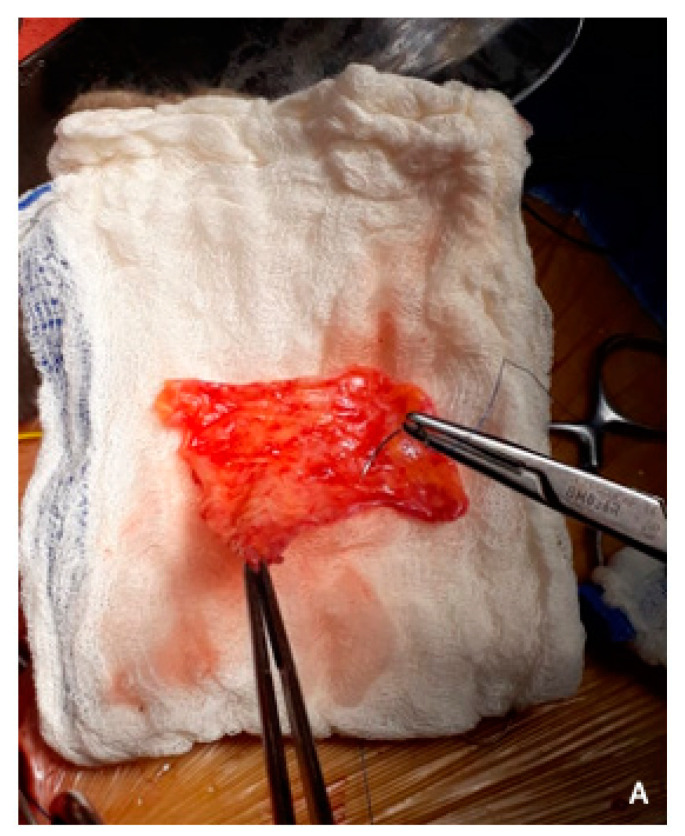
(**A**) Autologous parietal peritoneum was harvested from the right hypochondrium. (**B**) Type 2 vein reconstruction, using falciform ligament patch over the spleno–portal junction (white circle). The asterisk shows the distal pancreatic stump. The blue arrow shows the superior mesenteric artery. PV, portal vein. SMV, superior mesenteric vein.

**Figure 7 cancers-15-01509-f007:**
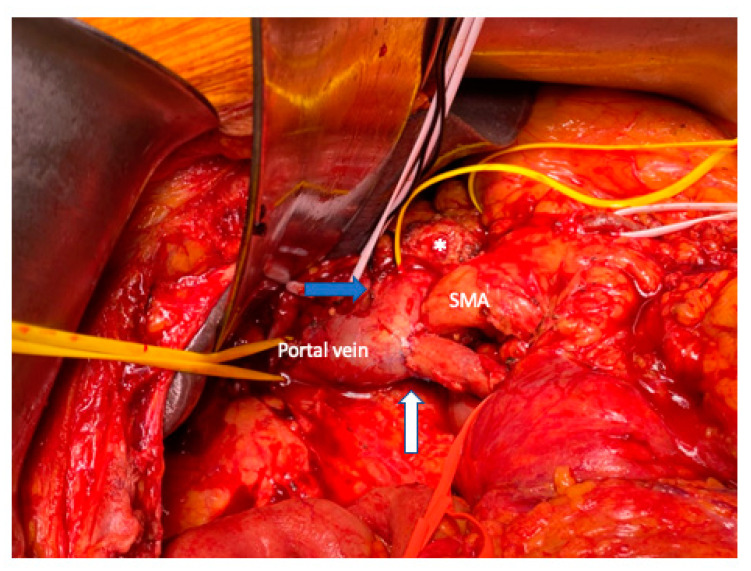
Type 3: segmental superior mesenteric vein resection (white arrow) with primary veno-venous anastomosis. The blue arrow shows the splenic vein. The asterisk, the distal pancreatic stump. SMA, superior mesenteric artery.

**Figure 8 cancers-15-01509-f008:**
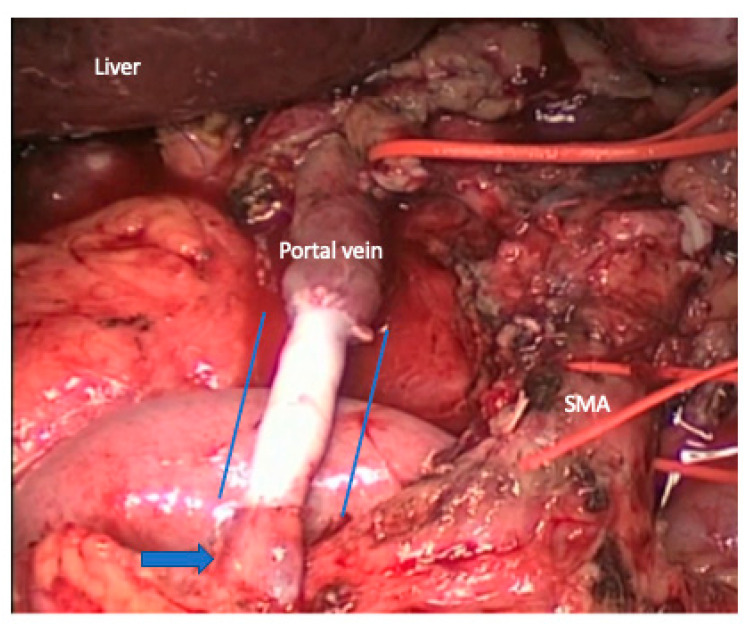
Type 4: segmental resection with interposed venous conduit. The spleno–portal confluent was complete resected. The blue arrow shows the superior mesenteric vein. The blue lines mark the cadaveric iliac vein interposed conduit of 6.5 cm long. SMA, superior mesenteric artery, which was clamped for 25 min to perform the venous anastomosis and avoid intestinal congestion.

**Figure 9 cancers-15-01509-f009:**
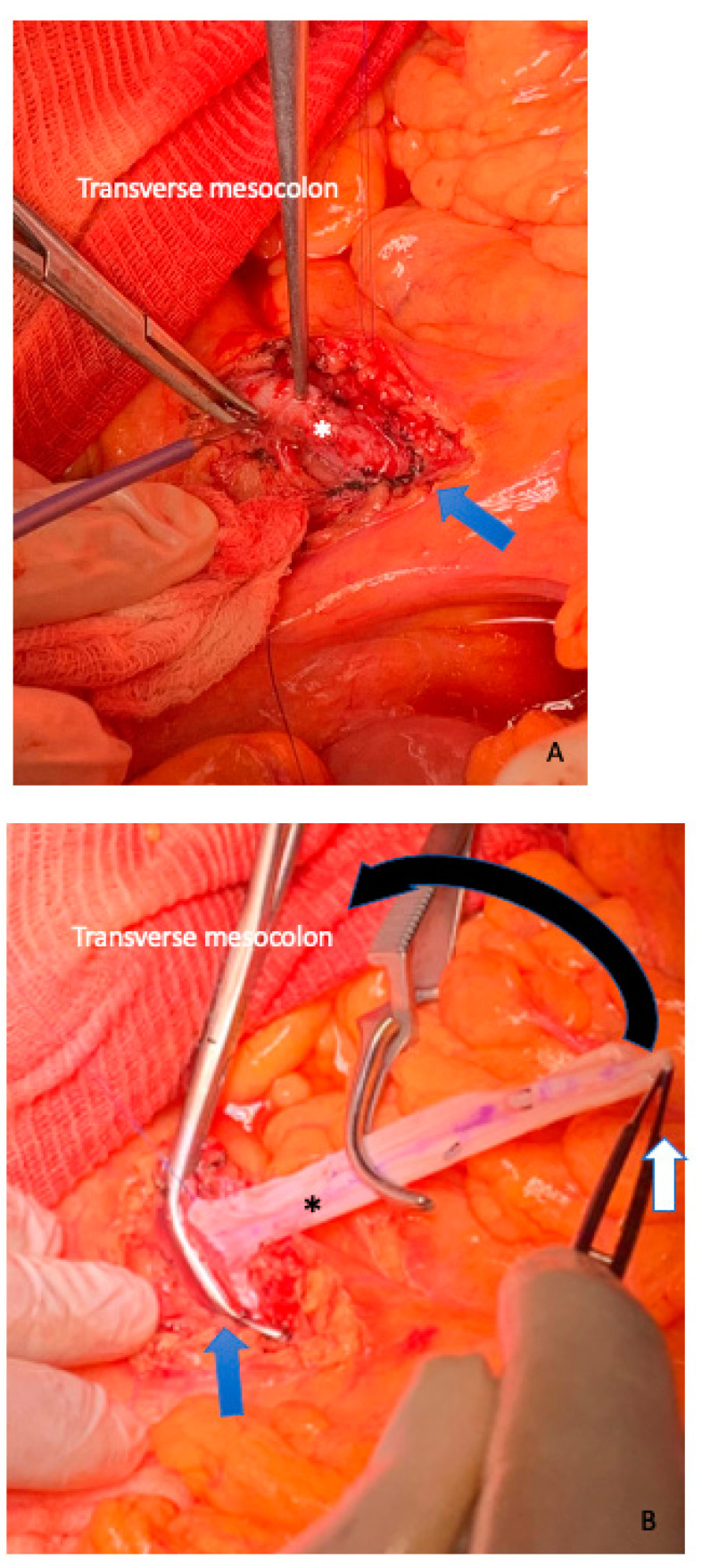
Venous jump graft technique. (**A**) Identification of the superior mesenteric vein (white asterisk) at the root of the mesentery (blue arrow). (**B**) Anastomosis between the superior mesenteric vein (blue arrow) and cadaveric iliac venous bank graft (black asterisk). The white arrow marks the distal segment that will be anastomosed with the portal vein or a collateral vessel. The jump graft is accessed through the transverse mesocolon (black arrow).

**Figure 10 cancers-15-01509-f010:**
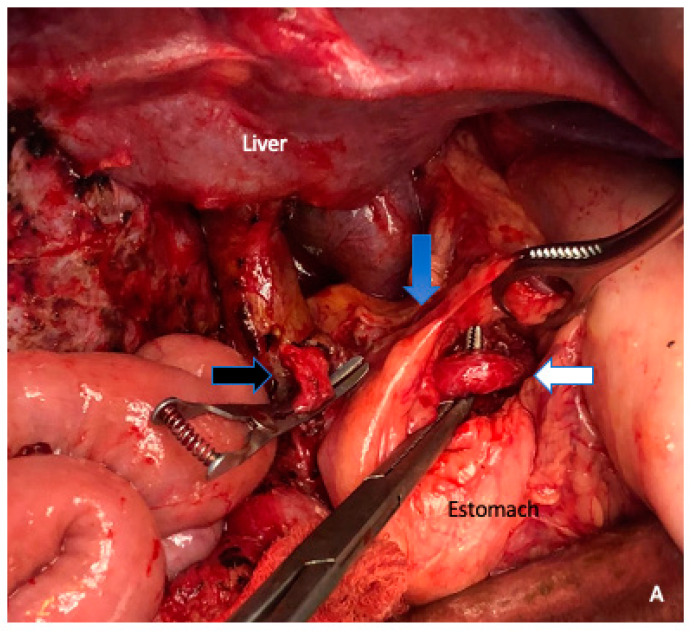
Splenic artery transposition technique with respective anastomosis with proper hepatic artery. (**A**) The right angle shows the splenic artery being dissected from its origin in the celiac trunk (blue arrow) distally (white arrow). The black arrow shows proper hepatic artery (bulldog clamp). (**B**) Section towards the distal portion of the splenic artery. The white arrow shows the distal end of the splenic artery that will be ligated. The blue arrow, the portion of the splenic artery that will be anastomosed to the proper hepatic artery (asterisk). The black arrow is marking the 180° rotation that the splenic artery will undergo to achieve a tension-free anastomosis. (**C**) Anastomosis between the splenic artery and proper hepatic artery. The white arrow is marking the clamp in the proximal segment of the splenic artery. The black arrow shows the 4 cm segment of the splenic artery that was rotated 180°, with the respective anastomosis to the proper hepatic artery (blue arrow).

**Figure 11 cancers-15-01509-f011:**
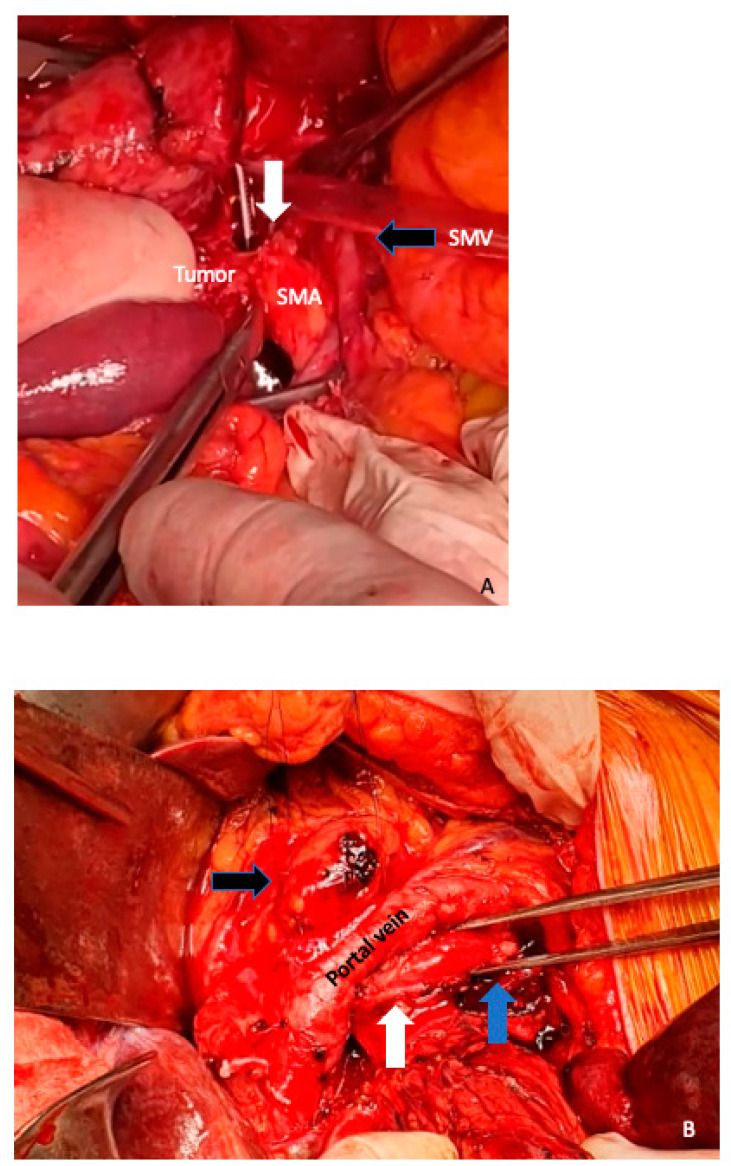
(**A**) Arterial divestment, is achieved by placing it in a plane between the uninvolved arterial wall and the tumor tissue of the affected arterial segment. The dissection plane should be placed between the periarterial nerve plexus and the arterial adventitia. We usually perform this maneuver with cold scissors to avoid thermal damage to the arterial wall, as can be seen in the image. The white arrow shows the limit between the tumor and the superior mesenteric artery (SMA) wall. Black arrow, retracted superior mesenteric vein (SMV) (**B**) The white arrow is showing the tunica adventitia of the superior mesenteric artery, which has then been removed distally, to give a tumor margin: the tunica media of the superior mesenteric artery can be seen at the level of the forceps and marked with a blue arrow. Pancreas stump (black arrow).

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
