# Peer review of "Technical Implications for Surgical Resection in Locally Advanced Pancreatic Cancer"

_cancers, 2023, doi:10.3390/cancers15051509_

Round 1

Reviewer 1 Report

I am honored to review this manuscript titled “Technical implications for surgical resections in locally advanced pancreatic cancer”. This review discussed the current surgical approaches for patients with LAPC. Many similar works have been done in this field although no data are from prospective clinical studies. I have the following comments:

·      This review lacks detail on complications and outcomes associated with those technical approaches. 

·      They concluded long-term outcomes after extended pancreatectomies are similar to standard resections and better than palliative care, although there is no level 1 evidence to support this conclusion. 

·      There are quite a few minor grammar/syntax/spelling errors. Please carefully edit the manuscript to eliminate these.

Author Response

February 13, 2023

Reference: Revised Manuscript cancers-2170607

Title: “Technical Implications for Surgical Resection in Locally Advance Pancreatic Cancer”

We thank the referees for their fair, thorough, and thoughtful review. Please see below our response to their comments. All the concerns raised by the reviewers have been addressed, and we hope that you will find the revised manuscript suitable for publication. The changes made in the manuscript were highlighted in yellow, with the added text in bold print and the deleted text crossed out. On behalf of the co-authors, I would like to thank the reviewers for their helpful and cogent comments.

All the authors have seen this version of the manuscript and agree with the modifications that have been performed.

Reviewers' comments:

Comment 1: This review lacks detail on complications and outcomes associated with those technical approaches.

Response: Thank you very much for the comment. We do not focus so much on these aspects, since the manuscript is, above all, technically oriented. Anyway, we added some of these aspects suggested by the reviewer in the different sections of the manuscript as follows:

  • Extended resection of neural and lymphatic tissue carries a risk of increased surgical morbidity, including adverse effects such as postoperative bleeding, uncontrolled diarrhea, and refractory ascites.
  • Some publications propose that the surgical morbidity is increased in extended resections while overall perioperative mortality appears to be similar compared with standard pancreatectomies [23]. This increased perioperative morbidity demands for close postoperative follow-up of these patients and an elaborate and aggressive management of complications that can only be provided in specialized high-volume centers.
  • Although several studies have suggested that pancreatic resections with PVR is associated with acceptable perioperative risk, performing venous resection adds undoubtedly a technical challenge to an already complex surgical procedure [24]. The benchmark cohort revealed a 4% in-hospital mortality, with a portal vein thrombosis rate ≤14% [24].

Comment 2:  They concluded long-term outcomes after extended pancreatectomies are similar to standard resections and better than palliative care, although there is no level 1 evidence to support this conclusion.

Response: Thank you very much for the comment. The conclusion was modified as follows: “Furthermore, through multimodal treatment systems and appropriate surgical resection techniques, the long-term outcome after extended pancreatectomies can be similar to standard resections and considerably better than palliative care despite the fact that we do not have the sufficient level of scientific evidence.”.

Comment 3: There are quite a few minor grammar/syntax/spelling errors. Please carefully edit the manuscript to eliminate these.

Response: Thank you for pointing this out. We had the manuscript reviewed by a native speaker and they were resolved.

Please do not hesitate to contact me if there is any further revision of our manuscript needed. Looking forward to a favorable response, I thank you in advance.

Sincerely,

Martin de Santibanes MD, PhD

Department of General Surgery. Hospital Italiano de Buenos Aires, Argentina

Juan D. Perón 4190. C1181ACH. Buenos Aires, Argentina.

Tel: +54-11 4981 4501

Fax: +54-11 4981 4041

E-mail: martin.desantibanes@hospitalitaliano.org.ar

Reviewer 2 Report

It is suggested that the authors choose some high-resolution photos to replace the photos in this manuscript, these unclear photos will affect the readers feeling.

Author Response

                                                                                                                February 13, 2023

Reference: Revised Manuscript cancers-2170607

Title: “Technical Implications for Surgical Resection in Locally Advance Pancreatic Cancer”

We thank the referees for their fair, thorough, and thoughtful review. Please see below our response to their comments. All the concerns raised by the reviewers have been addressed, and we hope that you will find the revised manuscript suitable for publication. The changes made in the manuscript were highlighted in yellow, with the added text in bold print and the deleted text crossed out. On behalf of the co-authors, I would like to thank the reviewers for their helpful and cogent comments.

All the authors have seen this version of the manuscript and agree with the modifications that have been performed.

Reviewers' comments:

I wanted to thank you for the positive feedback you gave to our work.

Comment 1: It is suggested that the authors choose some high-resolution photos to replace the photos in this manuscript, these unclear photos will affect the readers feeling.

Response: Thank you for pointing this out. Regarding the resolution of the photos. Perhaps you are looking at a compressed version of them. The photos are of the highest quality and in the format suggested by the journal. When I personally check them, they look great. We are going to correct this aspect with the editor

Please do not hesitate to contact me if there is any further revision of our manuscript needed. Looking forward to a favorable response, I thank you in advance.

Sincerely,

Martin de Santibanes MD, PhD

Department of General Surgery. Hospital Italiano de Buenos Aires, Argentina

Juan D. Perón 4190. C1181ACH. Buenos Aires, Argentina.

Tel: +54-11 4981 4501

Fax: +54-11 4981 4041

E-mail: martin.desantibanes@hospitalitaliano.org.ar

Reviewer 3 Report

First of all, I would like to congratulate the authors with their work, describing preoperative planning and intraoperative techniques and decision-making in locally advanced pancreatic cancer surgery. They have provided nice overview of their clinical experience, whereby referring to different approaches in the literature. I would like to make some major and minor comments that might help to further optimize this paper, enhancing its clinical applicability. Of note, I have made some suggestions for references as well, but please see this just as a suggestion. 

-Major comments- 

(1) First of all, I think that it will further improve the paper when the authors perform a (semi-)systematic review on surgical techniques in LAPC surgery. This will increase the clinical usefulness of the paper, whereby the authors and readers are able to compare this with their own techniques. 

(2) In my opinion, more attention should be paid to the importance of tumor biology instead of tumor anatomy when selecting patients for surgery, illustrated and described by the following paper: Oba et al. (2022) J Hepatobiliary Pancreat Sci

Important parameters for biology-based patient selection are serum CA19.9 and FDG-PET, who are shortly described by the authors. Although this is a technical paper, I think that this should be emphasized in a section on preoperatively planning. A useful literature on CA19.9 is Rose et al. (2020) Oncologist; A recent valuable literature for FDG-PET might be Abdelrahman et al. (2022) J Natl Compr Canc Netw.

(3) Repeatedly, the authors mention that preoperative imaging is key for surgical planning. Of course, this is true as recently underlined by the development of the Johns Hopkins LAPC score (Gemenetzis et al. 2022 Ann Surg Oncol). However, it would be great if the authors could emphasize more in detail the ‘failure’ of preoperative computed tomography to differentiate between fibrosis and vital tumor tissue: Park et al. (2021) Eur Radiol Nevertheless, there are various features on computed tomography that might be helpful for the preoperative surgical planning, such as the halo versus string sign (Javed et al. 2022 Curr Problm Diagn Radiol), potentially differentiating between the need for arterial resection versus divestment. Another interesting hypothesizing paper was recently published about the halo sign on contrast-enhanced MRI (Stoop et al. 2022 Langenbecks Arch Surg). 

(4) At last, it would be valuable if the authors could emphasize the complexity of preoperative and intraoperative multidisciplinary and surgical decision-making, requiring high-volume expertise in LAPC. Even if an arterial divestment seems sufficient, such operations should be performed in expert centers considering the always present risk for iatrogenic damage after which arterial resection is required. Useful studies to show the risk of iatrogenic damage requiring arterial resection (Stoop et al. 2022 Br J Surg) and the risk of complex arterial resection in centers with limited experience (Tee et al. 2018 J Am Coll Surg).

Minor comments:

(1) In the tiel, please correct the typo 'advance' into 'advanced'. 

(2) Have the authors considered to use the term 'induction therapy' instead of 'neoadjuvant therapy'? After all, neoadjuvant therapy applies to (borderline) resectable pancreatic cancer, whereas induction therapy applies to LAPC. 

(3) The authors describe in the introduction that R0 should be the aim of surgery. However, recent literature suggest that R0/R1 is not predictive anymore in BRPC/LAPC after preoperative chemo(radio)therapy: Klaiber et al. (2021) Ann Surg. Possibly, because of more adequate systemic disease control by multi-agent chemotherapy and thereby better biology-based patient selection for surgery. Another reason might be the non-reliable assessment of R status after chemotherapy, as described by Soer et al. (2021) J Gastrointest Oncol. What are the authors' thoughts on this? 

(4) In the section 'preoperative planning', the authors forget to mention BRPC in the spectrum of localized pancreatic cancer. 

(5) The authors refer to the 60% resection percentage of Hackert et al. 2016 Ann Surg. However, this also included M1 disease at time of diagnosis. To present a better, balanced percentage, it would be fair to use the resection percentages who are described in recent systematic reviews after FOLFIRINOX, gemcitabine-nab-paclitaxel, or in general: Brown et al. (2022) Br J Surg.

(6) A useful reference to illustrate the correlation between more extensive portomesenteric venous resection and worse outcome is the recent Dutch national series from Groen et al. (2022) Br J Surg.

(7) It might be useful to also describe something about the indication of total pancreatectomy in case of arterial resection with reconstruction. For instance, Loos et al. (2022) Ann Surg said that partial pancreatectomy with arterial resection does not increase the risk for mortality in comparison to total pancreatectomy. 

Author Response

February 16, 2023

Reference: Revised Manuscript cancers-2170607

Title: “Technical Implications for Surgical Resection in Locally Advance Pancreatic Cancer”

We thank the referees for their fair, thorough, and thoughtful review. Please see below our response to their comments. All the concerns raised by the reviewers have been addressed, and we hope that you will find the revised manuscript suitable for publication. The changes made in the manuscript were highlighted in yellow, with the added text in bold print and the deleted text crossed out. On behalf of the co-authors, I would like to thank the reviewers for their helpful and cogent comments.

All the authors have seen this version of the manuscript and agree with the modifications that have been performed.

Reviewers' comments:

First of all, I would like to congratulate the authors with their work, describing preoperative planning and intraoperative techniques and decision-making in locally advanced pancreatic cancer surgery. They have provided nice overview of their clinical experience, whereby referring to different approaches in the literature. I would like to make some major and minor comments that might help to further optimize this paper, enhancing its clinical applicability. Of note, I have made some suggestions for references as well, but please see this just as a suggestion.

-Major comments-

Comment 1: First of all, I think that it will further improve the paper when the authors perform a (semi-)systematic review on surgical techniques in LAPC surgery. This will increase the clinical usefulness of the paper, whereby the authors and readers are able to compare this with their own techniques.

Response: I want to thank you for the positive comment you have made to us in general terms of the manuscript. Regarding the suggestions of carrying out a (semi-)systematic review on surgical techniques in LAPC surgery, I agree that it would provide great methodological wealth, however, the objective of the manuscript is technical aspects and many of these surgical techniques are very recent, thus the short- and long-term results are in the process of being validated. So, there is a paucity of evidence sufficient even to definitively support them.

Comment 2: In my opinion, more attention should be paid to the importance of tumor biology instead of tumor anatomy when selecting patients for surgery, illustrated and described by the following paper: Oba et al. (2022) J Hepatobiliary Pancreat Sci

Response: Thanks for the suggestion and we agree on this point. We have added a comment about it as follows:

“A recent symposium of experts from Western and Eastern high-volume centers reported new resectability classifications from their respective institutions based on tumor biology, conditional status, pathology, and genetics, in addition to anatomical tumor involvement. Interestingly, experts from all the centers reached the agreement that anatomy alone is insufficient to define resectability in the current era of effective neoadjuvant therapy []”

Comment 3: Important parameters for biology-based patient selection are serum CA19.9 and FDG-PET, who are shortly described by the authors. Although this is a technical paper, I think that this should be emphasized in a section on preoperatively planning. A useful literature on CA19.9 is Rose et al. (2020) Oncologist; A recent valuable literature for FDG-PET might be Abdelrahman et al. (2022) J Natl Compr Canc Netw.

Response: All these suggestions have been introduced in the manuscript as follows:

Rose et al []  identified that the percent decrease in CA19-9 from baseline to minimum value (odds ratio [OR] 0.947, p ≤ .0001) and the percent increase from minimum value to final restaging CA19-9 (OR 1.030, p ≤ .0001) were predictive of tumor progression in patients with advanced pancreas cancer”.

“Recently, Abdelrahman et al []  showed that among patients with post-neoadjuvant therapy, FDG-PET highly predicts pathologic response (odds ratio, 43.2; 95% CI, 16.9-153.2), recurrence-free survival (hazard ratio, 0.37; 95% CI, 0.2-0.6), and overall survival (hazard ratio, 0.21; 95% CI, 0.1-0.4) superior to biochemical responses alone (CA 19-9)”.

Comment 4: Repeatedly, the authors mention that preoperative imaging is key for surgical planning. Of course, this is true as recently underlined by the development of the Johns Hopkins LAPC score (Gemenetzis et al. 2022 Ann Surg Oncol). However, it would be great if the authors could emphasize more in detail the ‘failure’ of preoperative computed tomography to differentiate between fibrosis and vital tumor tissue: Park et al. (2021) Eur Radiol Nevertheless, there are various features on computed tomography that might be helpful for the preoperative surgical planning, such as the halo versus string sign (Javed et al. 2022 Curr Problm Diagn Radiol), potentially differentiating between the need for arterial resection versus divestment. Another interesting hypothesizing paper was recently published about the halo sign on contrast-enhanced MRI (Stoop et al. 2022 Langenbecks Arch Surg).

Response: Thanks for this recommendation. All these suggestions have been introduced in the manuscript as follows:

“A recent development in post-process-rendering called cinematic rendering overcomes this by utilizing advanced light modeling to generate photorealistic 3D images with enhanced details. For local determination of resectability, vascular mapping allows for accurate assessment of major arteries and the portovenous system. For the portovenous anatomy it assists in determining the optimal surgical approach (extent of resection, appropriate technique for reconstruction, and need for mesocaval shunting). For arterial anatomy, vessel encasement either represents dissectible involvement via periadventitial dissection or true vessel invasion that is unresectable [] . Magnetic resonance imaging - halo sign, defined as replacement of solid perivascular (arterial and venous) tumor tissue by a zone of fatty-like signal intensity, might be helpful to assess the effect of induction chemotherapy in patients with LAPC [] . Further investigations incorporating quantitative parameters such as radiomics and deep learning may improve diagnostic performance of MDCT for predicting R0 resection  [].”

Comment 5: At last, it would be valuable if the authors could emphasize the complexity of preoperative and intraoperative multidisciplinary and surgical decision-making, requiring high-volume expertise in LAPC. Even if an arterial divestment seems sufficient, such operations should be performed in expert centers considering the always present risk for iatrogenic damage after which arterial resection is required. Useful studies to show the risk of iatrogenic damage requiring arterial resection (Stoop et al. 2022 Br J Surg) and the risk of complex arterial resection in centers with limited experience (Tee et al. 2018 J Am Coll Surg).

Response: These suggestions have been introduced in the manuscript as follows:

“Recently, Tee et al [] published the largest single-institution series specifically addressing indications, outcomes, and perioperative risk factors in pancreatectomy with AR. Despite having described a significant improvement in 90-day mortality over time; morbidity and the use of hospital resources remain unchanged. The most significant predictor of worse outcomes is post-pancreatectomy hemorrhage. Graft reconstruction and pancreatic fistula were also associated with increased major morbidity in their experience. It is highly recommended such cases be performed by surgeons with the specific anatomic comprehension and skillsets required not only to perform such complex resections, but also with the necessary institutional expertise and immediate availability of interventional radiology, complex endoscopy, and adequate intensive care facilities”.

Minor comments:

Comment: In the tiel, please correct the typo 'advance' into 'advanced'.

Response: This was corrected.

Comment: Have the authors considered to use the term 'induction therapy' instead of 'neoadjuvant therapy'? After all, neoadjuvant therapy applies to (borderline) resectable pancreatic cancer, whereas induction therapy applies to LAPC.

Response: Most publications use the term 'neoadjuvant therapy'. Thus, we prefer to keep this terminology.

Comment: The authors describe in the introduction that R0 should be the aim of surgery. However, recent literature suggest that R0/R1 is not predictive anymore in BRPC/LAPC after preoperative chemo(radio)therapy: Klaiber et al. (2021) Ann Surg. Possibly, because of more adequate systemic disease control by multi-agent chemotherapy and thereby better biology-based patient selection for surgery. Another reason might be the non-reliable assessment of R status after chemotherapy, as described by Soer et al. (2021) J Gastrointest Oncol. What are the authors' thoughts on this?

Response: We believe that a little more data is missing regarding the issue of surgical margins in LAPC surgical rescue. Possibly with time, we will have more precision in this regard.

Comment: In the section 'preoperative planning', the authors forget to mention BRPC in the spectrum of localized pancreatic cancer.

Response: This aspect is true. The idea was to define the poles of resectability and that generate less controversy regarding definition and subsequent therapeutic management.

Comment: The authors refer to the 60% resection percentage of Hackert et al. 2016 Ann Surg. However, this also included M1 disease at time of diagnosis. To present a better, balanced percentage, it would be fair to use the resection percentages who are described in recent systematic reviews after FOLFIRINOX, gemcitabine-nab-paclitaxel, or in general: Brown et al. (2022) Br J Surg.

 Comment: A useful reference to illustrate the correlation between more extensive portomesenteric venous resection and worse outcome is the recent Dutch national series from Groen et al. (2022) Br J Surg.

 Response: This suggestion has been introduced in the manuscript as follows:

A nationwide cohort analysis showed that patients with segmental resection, but not those who had wedge resection, had higher rates of major morbidity (odds ratio = 1.93, 95% CI 1.20 to 3.11) and worse overall survival (hazard ratio = 1.40, 95% CI 1.10 to 1.78), compared to patients without venous resection []”.

Comment: It might be useful to also describe something about the indication of total pancreatectomy in case of arterial resection with reconstruction. For instance, Loos et al. (2022) Ann Surg said that partial pancreatectomy with arterial resection does not increase the risk for mortality in comparison to total pancreatectomy.

Response: “Postpanreatectomy hemorrhage after pancreatectomy with AR may be a logical consequence of postoperative pancreatic fistula. To eliminate this risk, total pancreatectomy has been suggested []. However, a recent study found no protective effect of total pancreatectomy on its outcomes []”.

Please do not hesitate to contact me if there is any further revision of our manuscript needed. Looking forward to a favorable response, I thank you in advance.

Sincerely,

Martin de Santibanes MD, PhD

Department of General Surgery. Hospital Italiano de Buenos Aires, Argentina

Juan D. Perón 4190. C1181ACH. Buenos Aires, Argentina.

Tel: +54-11 4981 4501

Fax: +54-11 4981 4041

E-mail: martin.desantibanes@hospitalitaliano.org.ar

Round 2

Reviewer 1 Report

Revisions are adequate.

Author Response

Thank you for your positive feedback. 

Reviewer 3 Report

First of all, I would like to thank the authors' for their effort to process the reviewers' feedback. Hereby, I would like to give a point-wise response on a couple of the discussed points. 

> I still think that a systematic approach to collect literature on this topic would be beneficial, but I also understand the authors' point of view. Then, I suggest to focus some more on your personal philosophy next the surgical technical aspects. For instance, would it be interesting to describe how you are selecting your patients with LAPC for surgery after induction therapy? Which criteria based on anatomy, biology, and conditional parameters are you using? 

> In the section about preoperative surgical planning, I advise to state more specifically that vascular involvement is often times overestimated on CT imaging after chemotherapy, using the reference from Park and colleagues. I think that the paper from Habib et al. (2021) Surgery on the halo vs. string sign is also very useful to shortly mention, as this might be helpful for preoperative planning. 

> I understand that the authors would like to show the extremes by stating that a tumor is either resectable or locally advanced. However, it seems misleading/incomplete to me in lines 58-60, stating that a tumor is either resectable or locally advanced. Therefore, I still suggest to also mention borderline resectable pancreatic cancer. 

> I understand the authors' point of view about the R0/R1 status. However, even localized pancreatic cancer including LAPC should be approached as a systemic disease. That could be underlined more in the first section when talking about striving for R0. 

> When writing about the benchmarks for portomesenteric venous resections, I would say 4% or less. 

Author Response

February 22, 2023

Reference: Revised Manuscript cancers-2170607

Title: “Technical Implications for Surgical Resection in Locally Advanced Pancreatic Cancer”

We thank the referees for their fair, thorough, and thoughtful review. Please see below our response to their comments. All the concerns raised by the reviewers have been addressed, and we hope that you will find the revised manuscript suitable for publication. On behalf of the co-authors, I would like to thank the reviewers for their helpful and cogent comments.

All the authors have seen this version of the manuscript and agree with the modifications that have been performed.

Reviewers' comments:

First of all, I would like to thank the authors' for their effort to process the reviewers' feedback. Hereby, I would like to give a point-wise response on a couple of the discussed points.

Comment: I still think that a systematic approach to collect literature on this topic would be beneficial, but I also understand the authors' point of view. Then, I suggest to focus some more on your personal philosophy next the surgical technical aspects. For instance, would it be interesting to describe how you are selecting your patients with LAPC for surgery after induction therapy? Which criteria based on anatomy, biology, and conditional parameters are you using?

Response: Thanks for the suggestion, the criteria have not been drafted in such a precise way, since there are many parameters that are poorly defined in the literature and in practice up to no.  “We select our LAPC patients for surgery after neoadjuvant therapy, considering many of these anatomical and biologic criteria and conditional parameters described above, discussed on a case-by-case basis in our multidisciplinary committee”.

Comment: In the section about preoperative surgical planning, I advise to state more specifically that vascular involvement is often times overestimated on CT imaging after chemotherapy, using the reference from Park and colleagues. I think that the paper from Habib et al. (2021) Surgery on the halo vs. string sign is also very useful to shortly mention, as this might be helpful for preoperative planning.

Response: We consider that all these concepts that you suggested were introduced in the last revision, including the corresponded references. It had been expressed as follows. “Moreover, MDCT may underestimate the response of neoadjuvant therapy and there-fore the discrimination of the venous and/or arterial compromise since the discrimina-tion between fibrosis and viable tumor is very complex to date. A recent development in post-process-rendering called cinematic rendering overcomes this by utilizing ad-vanced light modeling to generate photorealistic 3D images with enhanced details. For local determination of resectability, vascular mapping allows for accurate assessment of major arteries and the portovenous system. For the portovenous anatomy it assists in determining the optimal surgical approach (extent of resection, appropriate technique for reconstruction, and need for mesocaval shunting). For arterial anatomy, vessel en-casement either represents dissectible involvement via periadventitial dissection or true vessel invasion that is unresectable [9]. Magnetic resonance imaging - halo sign, de-fined as replacement of solid perivascular (arterial and venous) tumor tissue by a zone of fatty-like signal intensity, might be helpful to assess the effect of induction chemo-therapy in patients with LAPC [10]. Further investigations incorporating quantitative parameters such as radiomics and deep learning may improve diagnostic performance of MDCT for predicting R0 resection [11]”.

Comment: I understand that the authors would like to show the extremes by stating that a tumor is either resectable or locally advanced. However, it seems misleading/incomplete to me in lines 58-60, stating that a tumor is either resectable or locally advanced. Therefore, I still suggest to also mention borderline resectable pancreatic cancer.

Response: It was added as follows, “The term borderline has been used to describe tumors that are potentially resectable, but that have some degree of vascular involvement. A borderline tumor would be one with reconstructable venous involvement (SMV or PV) and/or contact within 180° of the vascular circumferences of arterial structures”.

Comment: I understand the authors' point of view about the R0/R1 status. However, even localized pancreatic cancer including LAPC should be approached as a systemic disease. That could be underlined more in the first section when talking about striving for R0.

Response: Thank you very much for the comment. The concept was added in the introduction.

Comment: When writing about the benchmarks for portomesenteric venous resections, I would say 4% or less.

Response: It was corrected as suggested.

Please do not hesitate to contact me if there is any further revision of our manuscript needed. Looking forward to a favorable response, I thank you in advance.

Sincerely,

Martin de Santibanes MD, PhD

Department of General Surgery. Hospital Italiano de Buenos Aires, Argentina

Juan D. Perón 4190. C1181ACH. Buenos Aires, Argentina.

Tel: +54-11 4981 4501

Fax: +54-11 4981 4041

E-mail: martin.desantibanes@hospitalitaliano.org.ar